# Effect of Partial Substitution of Flour with Mealworm (*Tenebrio molitor* L.) Powder on Dough and Biscuit Properties

**DOI:** 10.3390/foods11142156

**Published:** 2022-07-20

**Authors:** Xinyuan Xie, Zhihe Yuan, Kai Fu, Jianhui An, Lingli Deng

**Affiliations:** Hubei Key Laboratory of Biological Resources Protection and Utilization, College of Biological Science and Technology, Hubei Minzu University, Enshi 445000, China; 202130346@hbmzu.edu.cn (X.X.); 072040208@hbmzu.edu.cn (Z.Y.); 071840403@hbmzu.edu.cn (K.F.); 2020049@hbmzu.edu.cn (J.A.)

**Keywords:** wheat dough, mealworm powder, pasting, farinograph, biscuit

## Abstract

Mealworm (*Tenebrio molitor* L.) is a type of edible insect rich in protein that has become popular as a protein-alternative ingredient in flour-based products to improve the nutritional properties of baking products. The mealworm powder substitution affected the pasting, farinograph, extensograph properties of wheat flour and the texture, nutritional, and sensory properties of the resulting soda biscuit. The pasting parameters (peak viscosity, trough viscosity, breakdown viscosity, final viscosity, and setback viscosity) and the water absorption decreased with the increased mealworm powder substitution level, which was ascribed to the dilution effect of mealworm powder. The farinograph parameters remained similar up to 15% substitution level. The extensograph results showed that mealworm powder substitution decreased the elastic properties of wheat dough as indicated by the consistently decreased extensibility, stretching energy, and stretching resistance, resulting in a significantly decreased baking expansion ratio of the soda biscuit. The protein, lipid, and dietary fiber content of the biscuits increased accordingly with the increased mealworm powder substitution level. The protein content of the soda biscuit was gradually increased from 9.13/100 g for the control (M0) to 16.0/100 g for that supplemented with 20% mealworm powder (M20), accompanied with the significantly increased essential amino acid content. Meanwhile, the fat and dietary fiber content of M20 exhibited 20.5 and 21.7% increase compared to those of M0. The score of the sensory attributes showed no significant difference up to 15% substitution level. The results demonstrated the 15% mealworm powder substitution level would not significantly affect the farinograph property, microstructure of wheat dough, and sensory acceptability.

## 1. Introduction

Edible insects have attracted both researcher and commercial attention as viable protein substitutes for animal products. Interest is particularly high because of the small environmental footprint related to insect farming, together with its high economic value and potential [1,2]. It has been estimated that, across the globe, at least 2 billion people depend on eating insects for their diet, with consumer interest in insect-based foods steadily increasing. In this regard, many different types of edible insects were successfully used for the production of fortified flour-based foods, including breads [3,4], cookies [5], pasta [6], and chips [7]. Insect industrialization is on the rise, showing great potential for application across a variety of sectors such as agriculture, forestry, and many others [8]. Among the variety of insects that are edible yellow mealworm (*Tenebrio molitor* L.) has grown in popularity, specifically in relation to industrial farming and food processing. In 2021, the European Food Safety Authority issued a positive scientific opinion on the safety of dried yellow mealworm (*T. molitor*) as a novel food, according to Regulation (EU) 2015/2283 [9]. Powder from the mealworm larva can provide up to 50% protein and up to 28% fats that include essential amino and fatty acids [4,10]. The use of mealworm larva powder in bakery products not only enriches baked goods with healthy proteins, but also improves the quality of the proteins as well as sensory properties of the goods. Severini et al. [11] added ground mealworm larva in amounts of up to 20/100 g to wheat dough for 3D-printed snacks. The resulting printed snacks were enriched with 10 and 20% amounts of ground insects and significantly increased the total essential amino acids in the snacks, from 32.5 g (0% insects) to 38.2 g (10% insects) and 41.3 g (20% insects)/100 g protein. Roncolini et al. [3] substituted mealworm (*T. molitor*) powder into bread doughs at 5 and 10% amounts in replace of wheat flour and found that the addition of mealworm powder did not negatively affect the technological features of either doughs or breads. Kowalski et al. [4] studied the effect of four kinds of insect flour on bread supplementation and found that 10% insect powder increases the amino acid score for lysine from 40% to almost 70%, compared to traditional wheat bread. These studies have revealed that edible insects, ground into powder such as mealworm powder, are both viable and beneficial options for use in bakery products.

Adding additional protein sources to bakery products not only affects the nutritional properties of the products but also greatly affects its processing properties. Additives in dough interfere with gluten, disturbing the formation of a gluten network and resulting in changes in rheology, subsequently affecting the quality of the product [12]. In the food industry, an improved understanding of farinograph properties of flour dough during processing is needed, especially in response to the relationship between properties and final product quality. Some studies have revealed that the rheology of wheat dough was affected by insect ingredients. Roncolini et al. [3] revealed that the development time of bread flour and dough stability were not affected by the addition of 5 and 10% mealworm powder. Roncolini et al. [13] found that a 10 and 30% substitution of lesser mealworm (*Alphitobius diaperinus*) powder decreased water absorption, dough development time and dough stability. Osimani et al. [14] assessed the blends of bread flour and cricket powder and found that a 10% substitution did not alter the mixing properties of the flour, whereas a 30% addition of cricket powder led to a higher dough development time and lower dough stability. Previous research has revealed that powders from different insect species can exert different effects on the rheology properties of wheat flour.

Owing to the low moisture content and long shelf life of biscuits, they differ from other bakery products [15,16]. Even though there is growing interest in biscuit consumption as a ready-to-serve caloric snack by the majority of the population, it lacks in nutrients [17]. Therefore, the demand for nutritionally improved biscuits has increased with the growing number of consumers who are inclined to follow healthy dietary patterns [18]. Among novel protein sources, mealworm powder has been a popular ingredient. Recently, Sriprablom et al. [5] added *Tenebrio molitor* and *Zophobas atratus* powders in wheat flour up to 30% supplementation for cookie making and found the nutritional values and the hardness of cookies significantly increased. However, little is still known about the effect of mealworm powder on the low-gluten wheat dough rheology (pasting, farinograph, and extensograph), which would affect the processing process and product properties of biscuit.

Hence, the study hypothesized that the substitution of mealworm powder for flour would not only affect the physical, nutritional and sensory properties of biscuits, but also affect the rheology of the wheat flour. The aim of this work was to investigate the effect partial substitution of low-gluten wheat flour with mealworm powder on the rheology properties of the dough and properties of soda biscuit. The pasting characteristics, farinographic, and extensograph properties were evaluated in order to analyze the rheology of the wheat dough formulated with mealworm powder. The physical properties, including color, microstructure, texture, baking expansion ratio of the soda biscuit were characterized. The proximate composition and amino acid composition were assessed to evaluate the nutritional improvement effect of mealworm substitution. The sensory evaluation was conducted by consumer acceptance test.

## 2. Materials and Methods

### 2.1. Materials

Low-gluten wheat flour with protein contents of 8.5% were purchased from Xinxiang Xinliang Cereals Processing Co., Ltd. (Xinxiang, Henan, China). Mealworm powder was purchased from Qingdao Sino Crown Biological Engineering Co., Ltd. (Qingdao, China). The salt, baking soda, butter, and yeast were purchased from a local supermarket.

### 2.2. Preparation of Wheat Flour Formulated with Mealworm Powder

The low-gluten wheat flour was substituted by mealworm powder at weight ratios of 0% (M0), 5% (M5), 10% (M10), 15% (M15), and 20% (M20), respectively.

### 2.3. Pasting Characteristics

The pasting properties of the wheat flour formulated with or without mealworm powder were tested by a rapid viscosity analyzer (RVA-Eritm, PerkinElmer, Waltham, MA, USA) according to American Association of Cereal Chemists (AACC) method 76–21 (AACC, 2000) [19].

### 2.4. Farinographic and Extensograph Property

As per the AACC Method 54–21 (AACC, 2000), a dough rheology test was performed using a farinograph (JFZD, Beijing Dongfu Jiuheng Instrument Technology Co., Ltd., Beijing, China) [20]. The elastic properties of dough formulated with mealworm powder were measured using a JMLD150 Extensograph (Dongfu, Beijing, China).

### 2.5. Biscuit Preparation

A schematic illustration of the biscuit preparation process is shown in Figure 1. Briefly, 2 g of salt and 1 g of baking soda were added into 150 g of the mixed wheat flour, followed by the addition of 3 g of yeast that was pre-dissolved in 60 g of milk. Then 30 g of melted butter was added into the dough, followed by kneading and fermentation at 30 °C for 30 min. Then the dough was shaped into thin sheets and cut into squares. The squares were pricked by a fork to create perforation and then baked for 14 min with up and down temperature of 165 and 145 °C.

### 2.6. Physiochemical Analysis

#### 2.6.1. Color

The color of the biscuit dough and biscuit formulated with mealworm powder was measured by a colormetric (CS-820N, Hangzhou CHNSpec Technology Co., Ltd., Hangzhou, China) on the basis of the *L**, *a**, *b** color system: *L** is the lightness, *a** goes from green to red and *b** goes from blue to yellow.

#### 2.6.2. Texture

The texture of the biscuit dough and biscuit were measured by a Texture Analyzer (TA-XT plus, Stable Micro Systems Ltd., Surrey, UK) equipped with P36R probe to 25% of the original sample height with a test speed of 2 mm/s under a Texture Profile Analysis (TPA) model. The assessed parameters were hardness, springiness, cohesiveness, gumminess and resilience for the dough and hardness, cohesiveness, chewiness and resilience for the biscuit, respectively [21].

#### 2.6.3. Baking Expansion Ratio

The baking expansion ratio of the biscuit was characterized by the change in the thickness of the biscuit before and after baking. The thickness of the biscuit was measured by a digital Vernier caliper.

#### 2.6.4. Scanning Electron Microscopy (SEM)

Wheat dough formulated with mealworm powder was freeze-dried and then sprayed with gold by ion sputtering under vacuum conditions. The dough and biscuit were observed with a scanning electron microscopy (TESCAN MIRA LMS, Brno–Kohoutovice, Brno, Czech Republic) with a secondary electron detector (SE) at an acceleration voltage of 15 kV.

#### 2.6.5. Proximate Compositions and Amino Acid Compositions

The proximate composition of mealworm powder was tested at the Pony Testing International Group (Beijing, China). The moisture, ash, protein, crude fat, and dietary fiber content are determined with reference to GB 5009.3-2016, GB 5009.4-2016, GB 5009.5-2016, GB 5009.6-2016, and GB 5009.88-2014 [22]. The carbohydrate content was calculated based on the sum of other contents. The amino acid compositions were analyzed according to the procedure of Son et al. [10] with modification. The tryptophan was not detected due to the acid hydrolysis.

### 2.7. Sensory Analysis

Soda biscuits were subjected to a sensory evaluation by 40 untrained panelists recruited from the university community [23]. Minimal information about the study was given to the panelists to reduce bias. Panelists consisted of 20 women and 20 men between the ages of 18 and 24 years. A consumer acceptance test was made on the biscuits using a nine-point hedonic scale form, where 1 indicated maximum dislike, 5 corresponded to neither like nor dislike, and 9 indicated maximum appreciation [24]. Sensory properties (appearance, odor, texture, taste, saltiness, and overall acceptability) were evaluated.

### 2.8. Statistical Analysis

The data were expressed as mean ± standard deviation and all the experiments were performed at least three times. The statistical analysis was performed using one-way analysis of variance (ANOVA) by Origin 8.0 software (OriginLab, Northampton, MA, USA). The significance level was set at *p* value  <  0.05.

## 3. Results and Discussion

### 3.1. Pasting Properties

RVA curves were analyzed and the pasting parameters were listed in Table 1. Values for peak viscosity, trough viscosity, breakdown viscosity, final viscosity, and setback viscosity were found to be highest for wheat flour without mealworm powder. The parameters showed a nearly linear decrease with increasing mealworm powder substitution gradients. Peak viscosity is a function of the extent of swelling of the starch granule [25]. A subsequent decrease of starch in the blend, together with the increase of mealworm powder, resulted in a decrease in peak viscosity. Breakdown indicates the stability of hot paste and lower breakdown suggests stronger resistance to the shear thinning effect of pastes [26]. The setback value reflects the increase in viscosity during the cooling stage, as starch granules cool and gelatinized starch undergoes retrogradation. Setback values of wheat flour formulated with mealworm powder were significantly lower than the control, indicating increased resistance to undesirable retrogradation [27]. The peak time also decreased with increased mealworm substitution gradients, indicating that the dilution effect on the gluten network was dominated [26]. Pasting temperature (PT) indicates the minimum temperature required to gelatinize the starch [28]. The PT of the wheat flour increased slightly when the mealworm powder substitution gradient exceeded 10%. The high fat and dietary fiber content of mealworm powder may alter the heat transfer of solids, resulting in an increasing trend for the pasting temperature [29].

### 3.2. Farinograph Properties

Dough mixing is a procedure in which flour and water are stirred until gluten is formed as a consequence of the increased interaction among dispersed and hydrated gluten-forming proteins. The analysis of farinograph properties can provide useful information about the effect of mealworm powder on water absorption and mixing characteristics of the dough [30]. As summarized in Table 2, the values of water absorption for low-gluten doughs with mealworm powder ranged from 55.67% (M5) to 52.50% (M20) and were significantly lower (*p* < 0.05) than that of dough without mealworm powder (60.97%). A similar decrease in water absorption was observed by Waseem et al. [31], wherein a significant decrease in water absorption was observed from 63% (whole wheat flour, 8.65% protein) to 55%, using a 20% spinach powder substitution. Fang et al. [32] reported a significantly decreased water absorption of wheat dough formulated with 10% isomaltodextrin (48.9%) compared to the control (63.0%), which is potentially attributed to the high dietary fiber content (>80%). Changes in the water absorption of flour are attributed to the presence of a large amount of fat and dietary fiber in the mixture [33]. The mealworm powder in this study contained 43.5% protein, 25.3% fat, 1.17% carbohydrate, 22.1% dietary fiber, 3.5% ash, and 4.43% water. The high content of fat and dietary fiber might retard the water absorption of wheat dough. On the contrary, research conducted on adding protein ingredients or hydrocolloids into wheat flour usually reported increased water absorption following substitution, due to the presence of a great number of hydroxyl groups forming hydrogen bonds with water. Yoon et al. [34] found that the soy protein concentrate (SPC) level increased water absorption from 54.83% to 79.27% after the wheat flour was replaced by SPC at 24/100 g. *Chlorella pyrenoidosa* powder (59.63% protein, 20.37 carbohydrate, 10.05% fat) increased the water absorption from 59.55% to 60.80% after a 3% substitution [25].

Dough development time (DDT), stability time (ST) and farinograph quality number (FQN) reflected the tightness of the gluten networks and the strength of the dough, with higher values suggesting stronger doughs [20]. As shown in Table 2, the addition of mealworm powder did not modify DDT of low-gluten dough to a large degree. A 20% substitution gradient of mealworm powder significantly increased stability time to 3.27 min when compared to that of pure low-gluten wheat flour (1.37 min). FQN is a measure of the ability of dough to retain its structure over time during mixing, which revealed a similar change as observed in ST values. The dough strength of the low-gluten was not significantly weakened by mealworm powder up to 15% substitution level [5].

Extensograph characteristics were used to elucidate the effect of mealworm powder on the viscoelastic properties of dough [35]. The total energy required for the dough to be stretched from the beginning through to breaking is represented by the stretching energy. As shown in Table 2, the stretching energy decreased together with an increase in mealworm powder ratio when compared to the control. Extensibility showed similar values up to 15% substitution gradients, with a decrease at 20% substitution. As extensibility (Ex) increases, dough tensile strength increases, meaning the dough becomes progressively easier to be stretched and consequently more resistant to breakage. Stretching resistance could reflect the gluten strength of the dough to a certain extent [25]. The stretching resistance and stretch ratio of M5 doughs were similar to pure low-gluten doughs; a decrease at higher substitution levels was observed.

The visual appearance of the biscuit dough and the biscuit before and after baking is shown in Figure 2. The dough without mealworm powder showed the highest value of lightness (L*) and then decreased with the increased mealworm powder substitution level. The redness (a*) increased accordingly due to the brown nature of the mealworm powder, while the yellowness (b*) showed no significant change after mealworm substitution. The 5% substitution level (M5) significantly increased hardness, springiness, cohesiveness, and gumminess (Table 3). The dough with 10% (M10) and 15% (M15) substitution level showed similar texture properties compared with the control (M0). The 20% substitution level resulted in decreased strength of wheat dough, as indicated by the decreased dough hardness compared to other samples. The effect of additives on dough rheology was dominated by gluten dilution effect or/and water competition mechanism. It has been reported that the lower water absorption corresponded to higher hardness [26]. When the mealworm substitution was low (5%), the continuity of the gluten network was not significantly affected, which was in accordance with the similar extensograph parameters. The water competition effect dominated the change in the dough, resulting in significantly higher hardness. When the substitution level increased, mealworm powder physically disrupted the continuity of the gluten network, the gluten dilution effect dominated with the higher ratio of mealworm powder, which was consistence with the decreased extensograph parameters. Roncolini et al. [13] also suggested that the mealworm powders might cause a formation of a less developed three-dimensional gluten network, resulting in a less viscoelastic dough, which was consistent with the results of extensograph properties. In order to understand the effect of mealworm powder within the gluten network on dough rheology, SEM images of the dough microstructure with a magnification of 1000× are shown in Figure 3. This analysis provides helpful information for understanding the interactions among mealworm powder, gluten proteins, and starch granules [35]. Continuous gluten networks containing starch granules and gluten films were observed in the dough without and with 5 and 10% mealworm powder. When the substitution level of mealworm powder reached 15%, the starch granules seemed wrapped and the gluten network was affected. This indicated that a high substitution level of mealworm powder affected the formation of the gluten network and the distribution of starch granules in the dough.

### 3.3. Physical Properties of Biscuit

Generally, the biscuit exhibited a darker color than their corresponding dough and their dark color also increased with increasing substitution level of insect powders, as shown in Figure 2. The yellowness (b*) increased significantly for the biscuit with mealworm powder compared to the control (M0). This could be due to an increasing amount of free amino acids and proteins enhancing the Maillard reaction, which takes place during baking between amine groups of amino acids and proteins and carbonyl compounds, e.g., reducing sugars [36]. The baking expansion ratio of the soda biscuit decreased significantly with the increased substitution level (Table 4). The control (M0) soda biscuit showed a baking expansion ratio of 239.4%, while that of M15 decreased to about 150%, indicating the mealworm substitution deteriorated the gas holding capacity. González et al. [37] found the specific volume of bread was significantly decreased by 5% mealworm powder substitution compared to the control. Khuenpet et al. [38] also reported the specific volume of bread significantly decreased with an increase in mealworm powder content. It can be concluded that the high substitution level of mealworm powder would retard the gluten formation, resulting in a decrease in gas retention ability during proofing and baking.

In this study, significantly decreased hardness of biscuit with a 20% substitution level (7967 N) was observed compared to the hardness of M0 (10,348 N). This could be attributed to the fact that the weaker gluten network created after 20% mealworm powder substitution, resulting in the softer texture of the soda biscuit. The cohesiveness, chewiness, and resilience increased significantly with the increasing mealworm substitution level from 5% to 15%. It has been reported that the biscuits prepared with ingredients with a higher protein content may have contributed strong binding of protein and starch by hydrogen bonding which occurred during dough development and baking [5,39].

### 3.4. Nutritional Values

The proximate compositions of the mealworm powder and soda biscuit are shown in Figure 4. The mealworm powder exhibited high content of protein (43.5/100 g) and lipid (25.3/100 g). The protein, lipid, and dietary fiber content increased accordingly with the increased mealworm substitution level. The energy of the M0, M5, M10, M15, and M20 were 1854, 1854, 1878, 1862, and 1911 kJ/100 g. Hence, the energy was not significantly increased up to the 15% substitution level, while the protein content was increased to 14.2/100 g for the M15 biscuit. The protein and fat content of the M20 biscuit was comparable to that of pork and beef [40]. The amino acid composition of the wheat flour, mealworm powder, and soda biscuits are listed in Table 5. The mealworm powder showed a high content of essential amino acids, which was consistent with previous research [10,11,41]. Accordingly, soda biscuits with mealworm substitution exhibited gradual increases in both essential and nonessential amino acids. Among the essential amino acids, threonine, valine, phenylalanine+tyrosine, and leucine exhibited a relatively high increased ratio in soda biscuits supplemented with 20% mealworm powder, which were 171, 138, 132, and 131%, respectively.

### 3.5. Sensory Evaluation

The sensory scores for the appearance, odor, texture, taste, saltiness, and overall acceptability are listed in Table 6. Overall, the mean scores for all sensory attributes and overall acceptability of soda biscuit showed no significant difference up to the 15% mealworm powder substitution level. The soda biscuit with 20% mealworm powder substitution showed a significantly decreased sensory score. For the biscuit to be considered acceptable using a 9-point hedonic scale, the samples must rate at least 5.0 score approval. Since the mean scores for all sensory attributes and overall acceptability of the soda biscuit with up to 15% mealworm powder substitution were higher than 5.0, their sensory properties were considered to be acceptable. Wendin et al. [42] also found the high acceptability of mealworm substitute crisps and pates. They found no significant difference in total liking between 10 and 30% addition of mealworm, nor between 0 and 10% addition, in any of the two products. According to an investigation of consumers’ attitude toward yellow mealworm chips (YMC) [7], consumers were interested in new YMC foods other than a disgusting accidental encounter with insects in food.

## 4. Conclusions

Given the results of the present study, it was deduced that mealworm substitution has an impact on the pasting, farinograph, and extensograph of wheat flour. Apart from the peak temperature, other pasting parameters decreased gradually due to the starch dilution effect of the mealworm powder. Water absorption decreased significantly with the increase of mealworm substitution gradients, due to the high content of fat and dietary fiber in the mealworm powder. After mealworm substitution, wheat dough becomes less elastic up to 15% substitution. The 20% mealworm substitution level showed an adverse effect on the dough strength and biscuit texture properties. The substitution of mealworm powder for flour improved the nutritional property by increasement of protein content and amino acid composition. The cohesiveness, chewiness, and resilience of the soda biscuit increased significantly with the increasing mealworm substitution level from 5% to 15%. The results of sensory evaluation suggested that the biscuits produced with insect powders up to 15% substitution level were sensorially acceptable.

## Figures and Tables

**Figure 1 foods-11-02156-f001:**
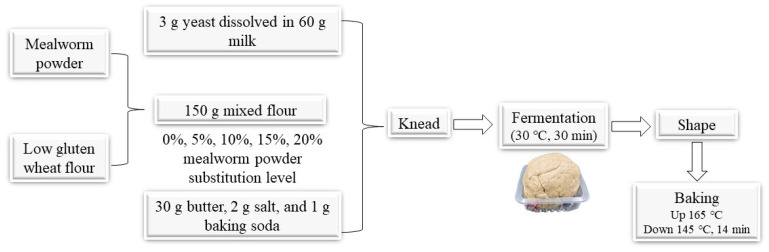
Schematic illustration of the soda biscuit preparation process.

**Figure 2 foods-11-02156-f002:**
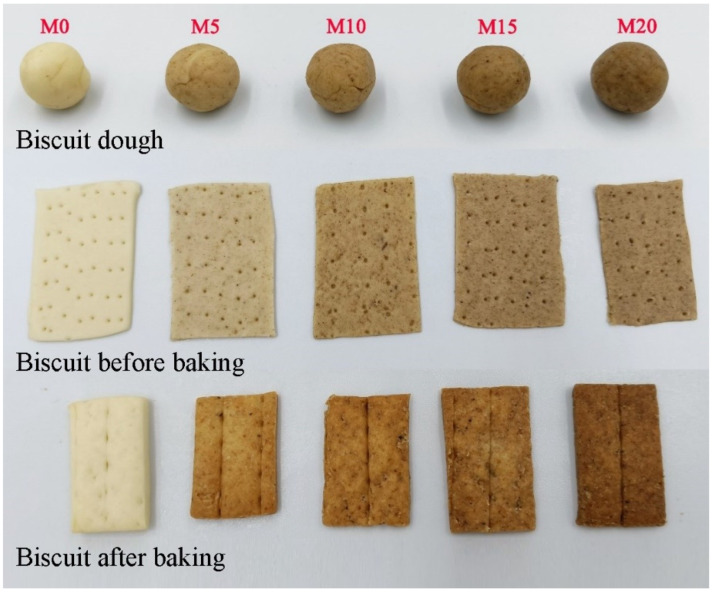
Visual appearance of the biscuit dough and the biscuit with various mealworm powder substitution levels before and after baking. The flour was substituted by mealworm powder at weight ratios of 0% (M0), 5% (M5), 10% (M10), 15% (M15), and 20% (M20), respectively.

**Figure 3 foods-11-02156-f003:**
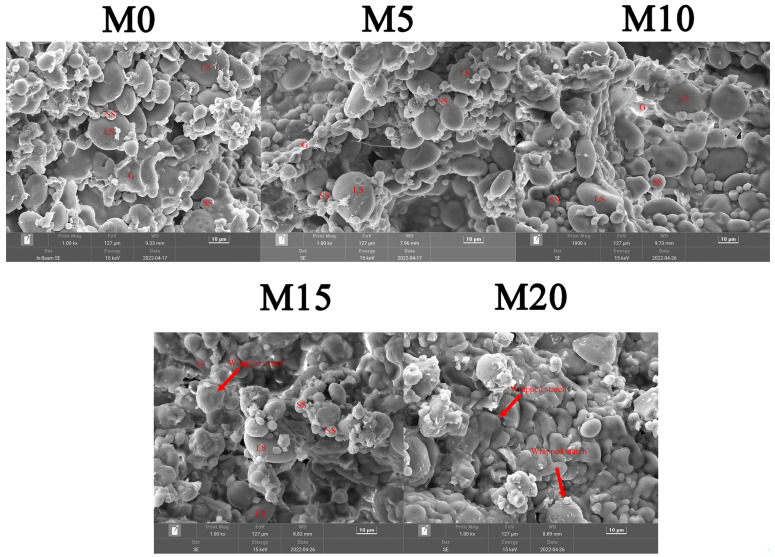
SEM images of the wheat dough with various mealworm powder substitution levels. LS, large starch; SS, small starch; G, gluten. The flour was substituted by mealworm powder at weight ratios of 0% (M0), 5% (M5), 10% (M10), 15% (M15), and 20% (M20), respectively.

**Figure 4 foods-11-02156-f004:**
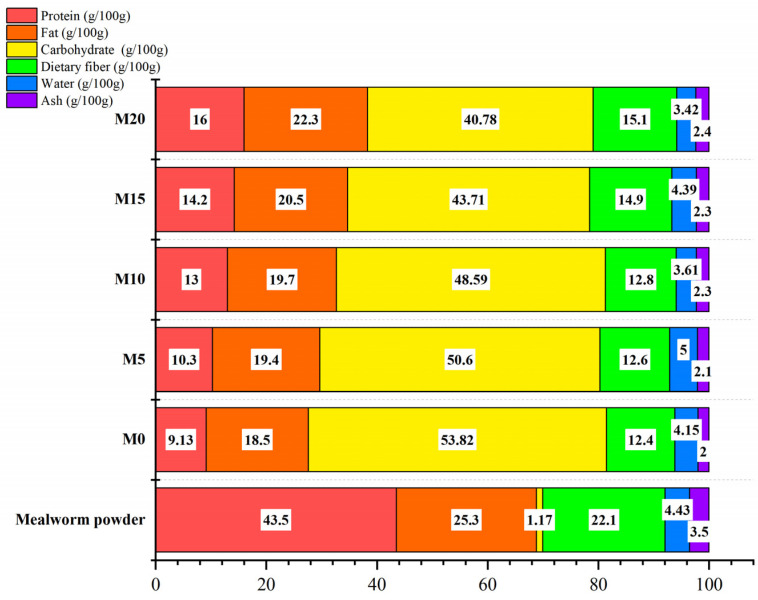
Proximate composition of the mealworm powder and the soda biscuit with various mealworm powder substitution levels.

**Table 1 foods-11-02156-t001:** The pasting characteristics of low-gluten wheat flour formulated with mealworm powder.

	Peak Viscosity (cp)	Trough (cp)	Breakdown (cp)	Final Viscosity (cp)	Setback (cp)	Peak Time (min)	Pasting Temperature (°C)
M0	1591 ± 2.65 ^a^	1087.67 ± 11.02 ^a^	503.33 ± 10.69 ^a^	2044.67 ± 28.73 ^a^	957.00 ± 18.33 ^a^	5.93 ± 0.01 ^a^	85.87 ± 0.08 ^b^
M5	1371.67 ± 9.50 ^b^	940 ± 13.45 ^b^	431.67 ± 9.24 ^b^	1775.67 ± 15.37 ^b^	835.67 ± 3.06 ^b^	5.80 ± 0.07 ^b^	85.30 ± 0.48 ^b^
M10	1180.33 ± 11.02 ^c^	811.33 ± 8.08 ^c^	369 ± 4.00 ^c^	1524.67 ± 20.01 ^c^	713.33 ± 12.01 ^c^	5.71 ± 0.03 ^b^	87.48 ± 0.03 ^a^
M15	1023 ± 8.66 ^d^	689 ± 5.2 ^d^	334 ± 3.46 ^d^	1286.67 ± 12.58 ^d^	597.67 ± 8.08 ^d^	5.47 ± 0.01 ^c^	87.5 ± 0.01 ^a^
M20	907 ± 14.4 ^e^	580.33 ± 20.55 ^e^	326.67 ± 7.09 ^d^	1071.33 ± 35.92 ^e^	491 ± 16.52 ^e^	5.29 ± 0.03 ^d^	87.48 ± 0.06 ^a^

Values in the same column followed by different superscripts are significantly different (*p* < 0.05).

**Table 2 foods-11-02156-t002:** The farinograph properties and extensograph properties of wheat flour substitute with mealworm powder.

	WA (%)	DDT (min)	ST (min)	DS (FU)	FQN	Stretching Energy (cm^2^)	Extensibility (mm)	Stretching Resistance (BU)	Stretch Ratio
M0	60.97 ± 0.26 ^a^	1.30 ± 0.08 ^b^	1.37 ± 0.12 ^b^	143.33 ± 8.18 ^a^	21.67 ± 1.25 ^b^	54.33 ± 2.49 ^a^	94.00 ± 2.94 ^a^	427.67 ± 5.91 ^a^	4.57 ± 0.12 ^a^
M5	55.67 ± 0.34 ^b^	1.20 ± 0.00 ^b^	1.53 ± 0.17 ^b^	113.33 ± 8.81 ^bc^	19.33 ± 1.70 ^b^	47.00 ± 1.63 ^b^	88.67 ± 1.25 ^ab^	403.67 ± 23.34 ^a^	4.57 ± 0.33 ^a^
M10	54.50 ± 0.43 ^c^	1.17 ± 0.05 ^b^	1.50 ± 0.16 ^b^	123.33 ± 11.26 ^ab^	19.33 ± 2.49 ^b^	33.00 ± 2.16 ^c^	92.67 ± 4.64 ^a^	268.00 ± 17.28 ^b^	2.90 ± 0.29 ^b^
M15	54.20 ± 0.14 ^c^	1.30 ± 0.08 ^b^	1.57 ± 0.12 ^b^	120.00 ± 3.56 ^abc^	20.00 ± 0.82 ^b^	26.67 ± 0.94 ^d^	87.00 ± 3.74 ^ab^	221.00 ± 13.06 ^b^	2.57 ± 0.26 ^b^
M20	52.50 ± 0.08 ^d^	2.60 ± 0.00 ^a^	3.27 ± 0.05 ^a^	95.67 ± 4.50 ^c^	56.67 ± 1.25 ^a^	27.67 ± 1.25 ^cd^	81.00 ± 3.56 ^b^	241.67 ± 2.05 ^b^	3.00 ± 0.14 ^b^

The flour was substituted by mealworm powder at weight ratios of 0% (M0), 5% (M5), 10% (M10), 15% (M15), and 20% (M20), respectively. WA, water absorption; DDT, dough development time; ST, stability time; DS, degree of softening; FQN, farinograph quality number. Values in the same column followed by different superscripts are significantly different (*p* < 0.053.3) biscuit dough properties.

**Table 3 foods-11-02156-t003:** The colorimetric and texture parameters of biscuit dough with various mealworm powder substitution levels.

	*L** (Lightness)	*a** (Redness)	*b** (Yellowness)	Hardness (N)	Springiness (mm)	Cohesiveness	Gumminess (N)	Resilience
M0	82.31 ± 1.87 ^a^	−0.21 ± 0.19 ^d^	18.68 ± 2.52 ^a^	1317.17 ± 185.42 ^b^	0.69 ± 0.04 ^b^	0.63 ± 0.03 ^ab^	833.24 ± 109.66 ^b^	0.13 ± 0.01 ^ab^
M5	72.66 ± 1.05 ^b^	3.70 ± 0.52 ^b^	22.07 ± 1.15 ^a^	1618.56 ± 87.33 ^a^	0.79 ± 0.06 ^a^	0.70 ± 0.08 ^a^	1140.48 ± 191.43 ^a^	0.16 ± 0.04 ^a^
M10	66.36 ± 3.58 ^c^	5.3 ± 0.45 ^c^	22.17 ± 1.94 ^a^	1469.10 ± 122.12 ^ab^	0.67 ± 0.05 ^b^	0.57 ± 0.04 ^b^	839.63 ± 125.82 ^b^	0.09 ± 0.01 ^bc^
M15	60.24 ± 2.17 ^d^	6.53 ± 0.61 ^a^	22.41 ± 2.12 ^a^	1487.66 ± 142.75 ^ab^	0.67 ± 0.02 ^b^	0.56 ± 0.05 ^b^	829.22 ± 108.96 ^b^	0.10 ± 0.01 ^bc^
M20	58.49 ± 1.54 ^e^	6.26 ± 0.33 ^a^	21.24 ± 0.94 ^a^	1293.90 ± 87.14 ^c^	0.66 ± 0.02 ^b^	0.53 ± 0.03 ^b^	693.17 ± 78.93 ^b^	0.08 ± 0.01 ^c^

The flour was substituted by mealworm powder at weight ratios of 0% (M0), 5% (M5), 10% (M10), 15% (M15), and 20% (M20), respectively. Values in the same column followed by different superscripts are significantly different (*p* < 0.05).

**Table 4 foods-11-02156-t004:** The colorimetric, baking expansion ratio, and texture parameters of the soda biscuit with various mealworm powder substitution levels.

	*L** (Lightness)	*a** (Redness)	*b** (Yellowness)	Baking Expansion Ratio/%	Hardness (N)	Cohesiveness	Chewiness	Resilience
M0	81.47 ± 1.52 ^a^	−0.41 ± 0.18 ^b^	22.0 ± 2.12 ^b^	239.4 ± 12.5 ^a^	10,348 ± 1662 ^a^	0.437 ± 0.084 ^b^	2516 ± 978 ^c^	0.327 ± 0.087 ^d^
M5	63.81 ± 3.02 ^b^	9.60 ± 2.17 ^a^	32.83 ± 1.79 ^a^	191.0 ± 9.6 ^b^	11,019 ± 2030 ^a^	0.544 ± 0.057 ^a^	3505 ± 1347 ^b^	0.438 ± 0.059 ^b^
M10	59.74 ± 0.87 ^b^	9.84 ± 0.94 ^a^	32.23 ± 1.06 ^a^	165.9 ± 1.8 ^c^	11,416 ± 2065 ^a^	0.599 ± 0.059 ^a^	3903 ± 1260 ^c^	0.516 ± 0.073 ^c^
M15	52.56 ± 4.11 ^b^	11.80 ± 3.12 ^a^	28.73 ± 1.57 ^c^	150.4 ± 7.3 ^cd^	10,925 ± 1657 ^a^	0.660 ± 0.049 ^a^	4093 ± 1124 ^a^	0.594 ± 0.070 ^a^
M20	50.63 ± 5.33 ^b^	11.43 ± 2.27 ^a^	28.27 ± 1.81 ^c^	147.5 ± 2.1 ^d^	7967 ± 1487 ^b^	0.558 ± 0.073 ^a^	2114 ± 740 ^c^	0.490 ± 0.087 ^bc^

The flour was substituted by mealworm powder at weight ratios of 0% (M0), 5% (M5), 10% (M10), 15% (M15), and 20% (M20), respectively. Values in the same column followed by different superscripts are significantly differen t (*p* < 0.05).

**Table 5 foods-11-02156-t005:** The amino acid composition of the wheat flour, mealworm powder and soda biscuit with various mealworm powder substitution levels.

g/100 g Sample (mg/g Protein)	Wheat Flour	Mealworm Powder	M0	M5	M10	M15	M20
Essential amino acids (EEA)							
His	0.30 (35.29)	2.41 (55.40)	0.34 (37.24)	0.36 (34.95)	0.46 (35.38)	0.52 (36.62)	0.62 (38.75)
Lys	0.14 (16.47)	2.92 (67.13)	0.16 (17.52)	0.26 (25.24)	0.35 (25.24)	0.45 (31.69)	0.54 (33.75)
Met + Cys	0.18 (21.18)	1.02 (23.45)	0.15 (16.43)	0.20 (19.42)	0.22 (16.92)	0.23 (16.20)	0.25 (15.63)
Phe + Tyr	0.65 (76.47)	5.97 (137.24)	0.64 (70.10)	0.77 (74.76)	0.98 (75.38)	1.23 (86.62)	1.49 (93.13)
Thr	0.21 (24.17)	2.17 (49.89)	0.21 (23.00)	0.32 (31.07)	0.39 (30.00)	0.51 (35.92)	0.57 (35.63)
Ile	0.29 (34.12)	2.67 (61.38)	0.29 (31.76)	0.36 (34.95)	0.40 (30.77)	0.57 (40.14)	0.67 (41.88)
Leu	0.56 (65.88)	3.73 (85.75)	0.55 (60.24)	0.71 (68.93)	0.77 (59.23)	0.94 (66.20)	1.07 (66.88)
Val	0.31 (36.47)	3.21 (73.79)	0.34 (37.24)	0.35 (33.98)	0.51 (39.23)	0.76 (53.52)	0.81 (50.63)
Nonessential amino acid							
Ala	0.47 (55.29)	5.55 (127.59)	0.29 (31.76)	0.51 (49.51)	0.74 (56.92)	1.04 (73.24)	1.18 (73.75)
Asp	0.31 (36.47)	4.04 (92.87)	0.25 (27.38)	0.54 (52.43)	0.63 (48.46)	0.87 (61.27)	1.02 (63.75)
Arg	0.32 (37.65)	2.75 (63.22)	0.20 (21.91)	0.35 (33.98)	0.40 (30.77)	0.54 (38.03)	0.63 (39.38)
Glu	2.69 (316.47)	6.34 (145.75)	1.66 (181.82)	2.40 (233.01)	2.47 (190.00)	2.76 (194.37)	2.87 (179.38)
Gly	0.30 (35.29)	2.53 (58.16)	0.25 (27.38)	0.38 (36.89)	0.46 (35.38)	0.61 (42.96)	0.69 (43.13)
Pro	1.11 (130.59)	3.79 (87.13)	0.80 (87.62)	1.25 (121.36)	1.35 (103.85)	1.29 (90.85)	1.49 (93.13)
Ser	0.31 (36.47)	2.17 (48.89)	0.28 (30.67)	0.40 (38.83)	0.47 (36.15)	0.58 (40.85)	0.65 (40.63)
Total EEA	2.64 (310.59)	24.1 (554.02)	2.68 (293.54)	3.33 (323.30)	4.08 (313.85)	5.21 (366.90)	6.02 (376.25)

**Table 6 foods-11-02156-t006:** The sensory scores of the soda biscuit with various mealworm powder substitution levels.

	Appearance	Odor	Texture	Taste	Saltiness	Overall Acceptability
M0	6.24 ± 1.20 ^a^	6.08 ± 1.15 ^a^	6.12 ± 1.01 ^a^	5.48 ± 1.12 ^a^	5.52 ± 1.36 ^a^	5.64 ± 1.03 ^a^
M5	5.68 ± 0.90 ^ab^	6.00 ± 1.00 ^a^	5.92 ± 1.03 ^ab^	5.44 ± 1.38 ^a^	5.60 ± 1.50 ^a^	5.65 ± 1.15 ^a^
M10	5.44 ± 1.08 ^ab^	5.60 ± 1.08 ^a^	5.76 ± 1.23 ^ab^	5.16 ± 1.59 ^a^	5.04 ± 1.56 ^ab^	5.09 ± 1.47 ^ab^
M15	5.56 ± 1.15 ^ab^	5.40 ± 1.25 ^a^	5.84 ± 1.07 ^ab^	4.88 ± 1.73 ^a^	5.12 ± 1.16 ^ab^	5.13 ± 1.28 ^ab^
M20	5.12 ± 1.09 ^b^	5.36 ± 0.81 ^a^	5.00 ± 1.50 ^b^	4.32 ± 1.67 ^a^	4.28 ± 1.56 ^b^	4.35 ± 1.49 ^b^

The flour was substituted by mealworm powder at weight ratios of 0% (M0), 5% (M5), 10% (M10), 15% (M15), and 20% (M20), respectively. Values in the same column followed by different superscripts are significantly different (*p* < 0.05).

## Data Availability

Data are not available in public datasets; please contact the authors.

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
