# Peer review of "Effect of Partial Substitution of Flour with Mealworm (Tenebrio molitor L.) Powder on Dough and Biscuit Properties"

_foods, 2022, doi:10.3390/foods11142156_

Round 1
Reviewer 1 Report
1 - Line 49 Please add space before [2]
2 - Line 52 Please add space before [3]
3 - Lines-62-65 Vegetarians and vegans completely exclude products of animal origin from their diet. So, this argument is incorrect.
4 - Line 73 Please add space before [2]
5 - Line 75 Please add space before [16]
6 - Line 77 Please add space before [17]
7 - Line 99 Please insert the city and country
8 - Line 105 Perten Instruments (Stockholm, Sweden) is now part of PerkinElmer, Waltham, Massachusetts, USA
9 - Line 106 Please insert “American Association of Cereal Chemists” before AACC
10 - Line 109 Please insert the city
11 - Line 125 Please insert the name of the manufacturer (Tescan), city and country
12 - Line 128 Please insert city and country
13 - Line 133 Please insert the name of the manufacturer, city and country
14 - Line 134 Please write “Texture Profile Analysis”, then “TPA”
15 - Lines 138-139 Please specify how the thickness of the biscuits was measured
16 - The information on 123-125 lines and 141-143 lines are almost identical. No need to repeat
17 - Line 146 Please delete the comma (…crude fat, and…)
18 - Line 150 Please add space before [9]
19 - Line 152 How did you determine the number of panelists? What was the gender and age structure of the panelists?
20 - Line 196 Please add space before [29]
21 - Line 208 Please add space before [32]
22 - Line 250 Please add space before [16]
23 - Line 254 Please specify in paper the working parameters for SEM analysis: WD, SEM HV, detector type, SEM magnification for each sample
24 - Line 323 Please add space before [40]
25 - Line 329-330 How do you explain the higher values for Appearance, Texture and Satiness presented by the M15 sample compared to M10 sample?
Author Response
Reviewer 1
1 - Line 49 Please add space before [2]
Response: We have revised it and checked the style problem in Endnote.
2 - Line 52 Please add space before [3]
Response: We have revised it.
3 - Lines-62-65 Vegetarians and vegans completely exclude products of animal origin from their diet. So, this argument is incorrect.
Response: Thanks for the notification, we have deleted the statement.
4 - Line 73 Please add space before [2]
Response: We have revised it.
5 - Line 75 Please add space before [16]
Response: We have revised it.
6 - Line 77 Please add space before [17]
Response: We have revised it.
7 - Line 99 Please insert the city and country
Response: We have added the city and country in Line 103.
8 - Line 105 Perten Instruments (Stockholm, Sweden) is now part of PerkinElmer, Waltham, Massachusetts, USA
Response: We have revised it accordingly in Line 110-111.
9 - Line 106 Please insert “American Association of Cereal Chemists” before AACC
Response: We have revised it accordingly in Line 111.
10 - Line 109 Please insert the city
Response: We have added the city in Line 115.
11 - Line 125 Please insert the name of the manufacturer (Tescan), city and country
Response: We have added the city and country in Line 148-149.
12 - Line 128 Please insert city and country
Response: We have added the city and country in Line 131.
13 - Line 133 Please insert the name of the manufacturer, city and country
Response: We have added the manufacture, city and country in Line 136.
14 - Line 134 Please write “Texture Profile Analysis”, then “TPA”
Response: We have revised it accordingly in Line 137.
15 - Lines 138-139 Please specify how the thickness of the biscuits was measured
Response: We have specified that the thickness of the biscuit were measured by a digital venier caliper in Line 143-144.
16 - The information on 123-125 lines and 141-143 lines are almost identical. No need to repeat
Response: Sorry for the mistake, we have deleted the information on the original Line 123-125.
17 - Line 146 Please delete the comma (…crude fat, and…)
Response: We have revised it accordingly in Line 153 and checked the whole manuscript.
18 - Line 150 Please add space before [9]
Response: We have revised it.
19 - Line 152 How did you determine the number of panelists? What was the gender and age structure of the panelists?
Response: The number of the panelists was referred to the study (Chauvin, M. A., Younce, F., Ross, C., & Swanson, B. (2008). Standard scales for crispness, crackliness and crunchiness in dry and wet foods: relationship with acoustical determinations. Journal of Texture Studies, 39(4), 345-368.). Panelists were recruited from the Hubei Minzu University students. Minimal information about the study was given to the panelists to reduce bias. Panelists consisted of 20 women and 20 men between the ages of 18 and 24 years.
20 - Line 196 Please add space before [29]
Response: We have revised it.
21 - Line 208 Please add space before [32]
Response: We have revised it.
22 - Line 250 Please add space before [16]
Response: We have revised it.
23 - Line 254 Please specify in paper the working parameters for SEM analysis: WD, SEM HV, detector type, SEM magnification for each sample
Response: We have added the working parameters in Material and Methods section (Line 149). The SEM observation was conducted with a secondary electron detector (SE) at an acceleration voltage of 15 kV. The SEM magnification of the samples were added in Line 267-268. The WD parameters varied for different samples and the detailed value can be found in the SEM image.
24 - Line 323 Please add space before [40]
Response: We have revised it.
25 - Line 329-330 How do you explain the higher values for Appearance, Texture and Saltiness presented by the M15 sample compared to M10 sample?
Response: Even though the average value of M15 showed slightly higher than M10, the significantly analysis revealed that there is no significantly difference between M10 and M15 in all the aspects of the sensory test. Hence, we stated in the manuscript that the mean scores for all sensory attributes and overall acceptability of soda biscuit showed no significantly difference up to 15% mealworm substitution level.

Reviewer 2 Report
Abstract - When I read the abstract, I don't see the reason behind adding mealworm to biscuits. Please address this. Also, in results mention about nutritional changes in biscuits after addition of mealworm.
Introduction - This section reads well. Please add a paragraph on structure of the paper. Also, following paper might be relevant to your work - Codesign of Food System and Circular Economy Approaches for the Development of Livestock Feeds from Insect Larvae
Material and Methods - Reads well
Results and Discussion - Figure 2 needs to clearly state before and after baking. I believe the middle layer is before and the bottom layer is after baking. Suggestion - can you add coloured Figure 3 rather than black and white
Conclusion - good
Overall, this paper discusses about substituting mealworm powder in biscuit dough with different variations. It does not highlight any novelty, its just trialling it with various compositions.
Author Response
Abstract - When I read the abstract, I don't see the reason behind adding mealworm to biscuits. Please address this. Also, in results mention about nutritional changes in biscuits after addition of mealworm.
Response: We have revised the abstract to address the background and the nutritional changes.
Introduction - This section reads well. Please add a paragraph on structure of the paper. Also, following paper might be relevant to your work - Codesign of Food System and Circular Economy Approaches for the Development of Livestock Feeds from Insect Larvae
Response: We have added a paragraph on the structure of the paper and reconstructed it to show the novelty of our study compared to the previous research. The reference was added accordingly.
Material and Methods - Reads well
Response: We have improved it according to other reviewers’ suggestions.
Results and Discussion - Figure 2 needs to clearly state before and after baking. I believe the middle layer is before and the bottom layer is after baking. Suggestion - can you add coloured Figure 3 rather than black and white
Response: We have revised Figure 2 to make it clear. For Figure 3, we have added the changes in the images with arrows.
Conclusion - good
Response: We have improved it according to other reviewers’ suggestions.
Overall, this paper discusses about substituting mealworm powder in biscuit dough with different variations. It does not highlight any novelty, its just trialling it with various compositions.
Response: We have addressed the novelty of our study in the Introduction section (Line 76-87).

Reviewer 3 Report
In this study mealworm powder is incorporated into biscuits formulation and its influences on the dough and biscuits is investigated. The manuscript is well written and topic is interesting however at least two papers with similar topic have been published in recent years. The authors should highlight the novelty of their work compared to those papers.
Other comments:
Line 15: delete “other” the
Line 63: Can vegetarians and vegans eat insect proteins?!
Lines 122-125: delete these sentences (It is written in section 2.6.4.)
Line 131: why the texture of biscuits is analyzed by TPA? This test is used for soft and flexible samples such as cakes, muffins, breads, etc. For biscuits usually Triple-point bend test is used.
Line 140: please add more details such as magnification, voltage, etc.
Line 210: Chlorella pyrenoidosa (Italic)
Line 213: write the full names for the first time and then use the abbreviations.
Line 239: why the hardness was increased at 5% substitution level and decreased at higher mealworm concentrations?
Figure 3: Show the changes in the images with arrows
Table 3: correct the order of letters. In some of the columns the highest value is shown by “a” letter while in other columns the lowest value is shown by “a” letter
Line 273: delete “of”
Line 284: I think “not” should be deleted because mealworm has retarded the formation of gluten network.
Table 4: correct the order of letters
Lines 293-295: please add a reference.
Line 343: change “cookies” to “biscuits”
Author Response
In this study mealworm powder is incorporated into biscuits formulation and its influences on the dough and biscuits is investigated. The manuscript is well written and topic is interesting however at least two papers with similar topic have been published in recent years. The authors should highlight the novelty of their work compared to those papers.
Response: We have addressed the novelty of our study in the Introduction section (Line 76-87).
Other comments:
Line 15: delete “other” the
Response: We have deleted it accordingly.
Line 63: Can vegetarians and vegans eat insect proteins?!
Response: Sorry for the mistake. Vegetarians and vegans completely exclude products of animal origin from their diet. So, this argument is incorrect. We have deleted the statement.
Lines 122-125: delete these sentences (It is written in section 2.6.4.)
Response: Sorry for the mistake, we have deleted the information in the original Line 122-125.
Line 131: why the texture of biscuits is analyzed by TPA? This test is used for soft and flexible samples such as cakes, muffins, breads, etc. For biscuits usually Triple-point bend test is used.
Response: Due to the limitation of the experimental conditions, the texture analyzer equipment was not equipped with a triple-point bending component. And the triple-point bending component is usually applied for testing fracturability. Hence, we referred to previous study who has also analyzed texture of biscuits by TPA (de Castro, Geovana Teixeira, et al. "Evaluation of the substitution of common flours for gluten‐free flours in cookies." Journal of Food Processing and Preservation 46.2 (2022): e16215).
Line 140: please add more details such as magnification, voltage, etc.
Response: We have added the details in Line 149 and Line 267-268.
Line 210: Chlorella pyrenoidosa (Italic)
Response: We have revised it accordingly in Line 217 and added the reference accordingly.
Line 213: write the full names for the first time and then use the abbreviations.
Response: We have revised it accordingly and checked the whole manuscript.
Line 239: why the hardness was increased at 5% substitution level and decreased at higher mealworm concentrations?
Response: We have added the analysis in Line 254-262. The effect of additives on dough rheology was dominated by gluten dilution effect or/and water competition mechanism. It has been reported that the lower water absorption corresponded to higher hardness. When the mealworm substitution was low (5%), the continuity of the gluten network was not significantly affected, which was in accordance with the similar extensograph parameters. The water competition effect dominated the change in the dough, resulting in significantly higher hardness. When the substitution level increased, mealworm powder physically disrupted the continuity of the gluten net-work, the gluten dilution effect dominated with the higher ratio of mealworm powder, which was consistence with the decreased extensograph parameters.
Figure 3: Show the changes in the images with arrows
Response: The large starch (LS), small starch (SS) and gluten (G) were labeled on the SEM image and the wrapped starch was shown with arrows.
Table 3: correct the order of letters. In some of the columns the highest value is shown by “a” letter while in other columns the lowest value is shown by “a” letter
Response: We have revised and made the highest value shown by “a” letter.
Line 273: delete “of”
Response: We have revised it accordingly.
Line 284: I think “not” should be deleted because mealworm has retarded the formation of the gluten network.
Response: Sorry for the mistake and we have revised it accordingly.
Table 4: correct the order of letters
Response: We have revised and made the highest value shown by “a” letter.
Lines 293-295: please add a reference.
Response: We have added the references accordingly.
Line 343: change “cookies” to “biscuits”
Response: We have revised it accordingly in Line 360

Round 2
Reviewer 2 Report
The authors have addressed all my concerns.
Reviewer 3 Report
The manuscript is acceptable